# Incidence and Risk Factors of Worsened Activities of Daily Living Status Three Months after Intensive Care Unit Discharge among Critically Ill Patients: A Prospective Cohort Study

**DOI:** 10.3390/jcm11071990

**Published:** 2022-04-02

**Authors:** Kyohei Miyamoto, Mami Shibata, Nozomu Shima, Tsuyoshi Nakashima, Rikako Tanaka, Keita Nakamoto, Yuriko Imanaka, Seiya Kato

**Affiliations:** 1Department of Emergency and Critical Care Medicine, Wakayama Medical University, Wakayama 641-0012, Japan; chicchilion.208@gmail.com (M.S.); nozowmu@gmail.com (N.S.); nakanakamizumizu@gmail.com (T.N.); boudanpam@yahoo.co.jp (R.T.); katos@wakayama-med.ac.jp (S.K.); 2Department of Nursing, Wakayama Medical University Hospital, Wakayama 641-0012, Japan; k11554212000@yahoo.co.jp (K.N.); 0322.aikojunkie@gmail.com (Y.I.)

**Keywords:** activity of daily living status, disability, risk factors, post-intensive care syndrome

## Abstract

Background: We aimed to determine risk factors associated with worsened activity of daily living (ADL) status three months after intensive care unit (ICU) discharge. Methods: In this prospective, observational study, we enrolled critically ill adult patients that were emergently admitted to an ICU. We assessed ADL status by Barthel index score prior to ICU admission and three months after ICU discharge. The primary outcome was worsened ADL status, defined as a ≥10 decrease in Barthel index score. Results: We enrolled 102 patients (median age was 72 years old, 55% were male, and 87% received mechanical ventilation during ICU stay), and 42 patients (41%) had worsened ADL status three months after discharge from ICU. Multivariate analysis revealed that older age (>70 years old; adjusted odds ratio (aOR) 3.68; 95% confidence interval (95%CI) 1.33–10.19), high burden of chronic illness (aOR 4.11; 95%CI 1.43–11.81), and longer duration of mechanical ventilation (≥4 days; aOR 2.83; 95%CI 1.04–7.69) were independent risk factors for worsened ADL status at three months. Conclusions: Almost half of the critically ill adult patients in this cohort had worsened ADL status after ICU discharge. Older age, high burden of chronic illness, and longer duration of mechanical ventilation were risk factors for worsened ADL status.

## 1. Introduction

After recovery from critical illness, patients frequently have new or worsened long-term functional physical, cognitive, and mental problems, a phenomenon known as post-intensive care syndrome (PICS) [1]. Physical and cognitive impairment often leads to functional disability, such as adversely affected activity of daily living (ADL) status. Approximately 30% of survivors from critical illness were reported to have reduced ADL status [2,3]. Early prediction of impairment of ADL status after intensive care unit (ICU) discharge may facilitate the implementation of preventive measures and help patients and their families better anticipate the post-ICU course.

PICS has various components, each of which with different risk factors. For example, well-known risk factors of physical impairment include longer duration of mechanical ventilation or systemic inflammatory response, and those of cognitive impairment include delirium in ICU [4].

Risk factors of worsened ADL status include older age and/or muscle weakness at hospital discharge, but the literature has been inconclusive [5,6]. One factor in this is the baseline information; some studies evaluated the baseline functional status in the same way for the evaluation at post-ICU follow-up, while some did not [2,5,6]. For the identification of risk factors, an exact evaluation of baseline status seems to be important. A second factor for being inconclusive about risk factors is the variables included in the multivariate model in each study; some studies included variables during ICU stay (e.g., organ support) in their multivariate model, while others included mainly baseline variables, and this might influence the results [5,7].

By evaluating variables during ICU stay, we could find the modifiable risk factors for implementing preventive measures. We, therefore, conducted this study to elucidate exact and modifiable risk factors of worsened ADL status using a prospective cohort including information on baseline ADL status and variables during ICU stay.

## 2. Materials and Methods

### 2.1. Study Design and Participants

This study is a predefined sub-analysis of the Wakayama Post-Intensive Care Syndrome (W-PICS) study, which is a single-center prospective observational study. It was approved by the Wakayama Medical University Institutional Review Board (approval number 1864) and registered to the UMIN Clinical Trial Registry on 1 September 2016 (registration no. UMIN000023743, https://upload.umin.ac.jp/cgi-open-bin/ctr/ctr_view.cgi?recptno=R000027346 (accessed on 3 March 2022)). Written informed consent was obtained from all patients or their proxies before enrollment.

In the W-PICS study, we enrolled 204 critically ill adult patients that were emergently admitted to our mixed ICU between September 2016 and August 2018. We excluded patients who were expected to die or discharge from ICU within 48 h of admission. The original purpose of the W-PICS study was to describe the epidemiology of PICS symptoms in Japan, and we evaluated the prevalence of reduced basic activity of daily living (ADL) status using the Barthel index (BI), and psychiatric symptoms, such as anxiety, depression, and post-traumatic stress disorder symptoms using Hospital Anxiety and Depression Scale (HADS) and the Impact of Event Scale-Revised (IES-R) [8,9,10]. We evaluated these symptoms by questionnaires completed by patients or their proxies 3 and 12 months after ICU discharge. We also evaluated BI scores before ICU admission (baseline) from information gained from patients’ proxies. The main results of the W-PICS study have been previously reported elsewhere [3]. The present sub-analysis includes all patients (no missing data) of BI both before ICU admission and three months after ICU discharge, and who can be evaluated to ascertain whether there was worsened ADL status between before and after their admission to ICU.

### 2.2. Outcomes

The primary outcome of the present study was worsened basic ADL status three months after ICU discharge, which was defined as a ≥10 decrease in BI score (deterioration) from the BI score before ICU admission (baseline) to that at three months. BI scores range from 0 (totally dependent) to 100 (fully independent) [8]. For the definition of the primary outcome, we applied the cutoff of 10 points of the change in BI, because the minimal clinically important difference in BI was reported to be 9.25 points [11].

### 2.3. Statistical Analysis

Continuous variables are presented as the median and interquartile range (IQR) or mean ± standard deviation. Categorical variables are presented as numbers and percentages. To compare the two groups, we used the Wilcoxon rank-sum test or *t*-test for continuous variables as appropriate, and Fisher’s exact test for categorical variables. To find the predictive factors associated with the primary outcome, we used univariate and multivariate logistic regression models. For variables included in logistic regression models, continuous variables which did not follow normal distribution were changed to binary variables according to their median values or previously reported cutoff values. For example, we used median values for age, and length of mechanical ventilation or ICU stay, and we used previously reported cutoff values for ADL disability at baseline (BI ≤ 60) [12]. Variables with *p* < 0.20 in univariate models were used in the multivariate logistic regression model. All statistical tests were two-sided, and *p* values < 0.05 were considered statistically significant. For all analyses, we used JMP Pro version 13.0 (SAS Institute, Cary, NC, USA).

## 3. Results

From the cohort of the W-PICS study including 204 patients, we enrolled 102 patients without missing BI data at baseline and 3 months after ICU discharge (Figure 1). Patient characteristics are shown in Table 1. The median age was 72 years, and 56 patients were male (55%). A major reason for ICU admission was sepsis (*n* = 42; 41%), followed by trauma (*n* = 19; 19%). The median and IQR score of BI prior to ICU admission was 100 (79–100), and those at three months were 90 (40–100). In total, 42 patients (41%) had worsened basic ADL status. Patient characteristics included being older and with a lower BI score prior to ICU admission, and receiving a longer duration of mechanical ventilation during the stay in ICU.

Table 2 shows predictors for worsened ADL status between baseline and at three months. In univariate logistic regression analysis, variables associated with worsened ADL status were age >70 years (odds ratio (OR) 3.35; 95% confidence interval (CI) 1.45–7.70), high burden of chronic illness (Charlson comorbidity index >1, OR 3.62; 95%CI 1.53–8.56), and ADL disability prior to ICU admission (OR 3.03; 95%CI 1.08–8.52). In multivariate logistic regression analysis, we found three independent risk factors: age > 70 years (adjusted OR 3.68; 95%CI 1.33–10.19), high burden of chronic illness (adjusted OR 4.11; 95%CI 1.43–11.81), and ≥4 days of mechanical ventilation received during the stay in ICU (adjusted OR 2.83; 95%CI 1.04–7.69).

Number of patients with 0, 1, 2, and 3 risk factors are 18, 42, 34, and 8, respectively. And 1 (6%), 14 (33%), 21 (62%), and 6 (75%) of whom experienced worsened ADL status at three months. Figure 2 shows the trajectories of basic ADL status during the year after ICU discharge among 57 patients with all the BI scores available at baseline, and 3 months and 12 months after ICU discharge. Among 18 patients who experienced worsened ADL status at 3 months, 12 patients (67%) subsequently experienced an improvement in basic ADL status at 12 months. Among the remaining 39 patients, 36 (93%) maintained the level of basic ADL status.

## 4. Discussion

In the present study, 41% of critically ill adult patients had worsened ADL status three months after ICU discharge in the mixed ICU setting. Three independent risk factors were identified: older age (>70 years), high burden of chronic illness (Charlson comorbidity index >1), and longer duration of mechanical ventilation during the patient’s stay in ICU (≥4 days). About three-quarters of patients who had all risk factors eventually developed worsened ADL status, and less than one-tenth of patients who had none of the risk factors developed worsened ADL status. About two-thirds of patients with worsened ADL status at three months subsequently experienced improvement of ADL status during the year after discharge from ICU.

Older age was strongly associated with long-term functional disability among critically ill patients in several previous studies [2,6]. A previous prospective observational study that enrolled 821 patients with respiratory failure or shock reported that 32% of patients experienced at least partial basic ADL dependency (measured by Katz ADL) three months after ICU stay [2]. Older age was also reported to be independently associated with reduced basic ADL status at three months [2]. Another prospective observational study that enrolled critically ill patients that underwent mechanical ventilation reported that 148 out of 406 patients alive at 6 months (36%) experienced new-onset disability measured by WHODAS 2.0, and older age was again an independent risk factor of death or new disability [6]. Similarly, another prospective observational study using BI also reported that older age was independently associated with worsened ADL status [5]. Concordant with previous studies, our study also found older age was a risk factor for worsened ADL status. Irrespective of the measurement of functional status used, older age is a strong risk factor in predicting worsened functional status. Similarly, chronic illness is reported to be an independent risk factor for long-term functional disability, which is also concordant with our results [13].

Other than older age and the high burden of chronic illness, the abovementioned studies reported several risk factors associated with disabled functional status: disease severity, the reason for ICU admission, and muscle weakness at hospital discharge [5,6]. The literature that includes variables obtained during ICU stay in multivariate models is scarce, so modifiable risk factors of worsened functional status remain unclear. Interestingly, a previous study using a sepsis cohort from randomized control trials reported that a longer duration of organ support (e.g., ventilator or dialysis) was independently associated with impaired health-related quality of life (HRQoL) related to mobility and self-care [7]. Specifically, this concerns HRQoL and not a functional disability, but considering the results of our study in which a longer duration of mechanical ventilation was a risk factor for worsened ADL status, we can speculate that a longer duration of organ support might be a risk factor of worsened functional status.

By using the three risk factors identified in our study, we can roughly estimate the probability of worsened ADL status three months after ICU discharge. Even if ADL status worsened three months after ICU discharge, more than half of patients would subsequently regain function to some extent during one year. Sharing the prediction and estimated trajectory of ADL status with patients and their families would contribute to better decision making during their ICU stay.

A longer duration of mechanical ventilation was identified as a modifiable risk factor. Worsened ADL status could be prevented by implementing measures for early weaning from mechanical ventilation, such as the ABCDEF bundle [14]. Chronic critical illness (persistent organ dysfunction) was reported to be associated with worse physical function and HRQoL twelve months after ICU discharge, so early liberation from organ dysfunction other than respiratory failure might also be important to reduce the burden of disabled ADL status [15].

Our study has several limitations. First, while it evaluated the patients’ basic ADL status, their HRQoL was not evaluated. In critically ill patients, basic ADL status measured by BI was reported to be correlated with their HRQoL [16]. However, the well-known “disability paradox” must also be considered—some severely disabled patients adapt to their circumstances and feel greater happiness and quality of life than healthy people [17]. Decisions on treatment should not be made merely based on the prediction of worse functional status. As a second limitation, we considered the cutoff of ≥10 change in BI score to be clinically relevant, but this was derived from the previously reported minimal clinically important difference in patients with strokes, not in ICU survivors [11]. Future studies should validate the minimal clinically important difference in BI score in ICU survivors. A third limitation is that this study excluded many patients from the primary analysis, mainly due to their death within 3 months and missing BI data. The substantial number of patients lost to follow-up during a prolonged research period is a common problem in PICS research. However, missing BI data in our study were relatively minimal, because the response rate to the questionnaires was sufficiently high (80%), and proxies, as well as the patients themselves, could complete the BI questionnaires. As a final limitation, this was a single-center study, and future multicenter studies should aim to validate its results. Despite these limitations, the results of our study are almost concordant with the results of the above-mentioned previous studies, which may assure the robustness of our results.

## 5. Conclusions

Among critically ill adult patients, almost half experienced worsened ADL status three months after ICU discharge. Independent risk factors for worsened ADL status were older age (>70 years), high burden of chronic illness (Charlson comorbidity index >1), and longer duration of mechanical ventilation (≥4 days). Worsened ADL status at three months subsequently improved within one year in the majority of the patients. Sharing the prediction and trajectory of ADL status with patients and their families could contribute to better decision making during their ICU stay.

## Figures and Tables

**Figure 1 jcm-11-01990-f001:**
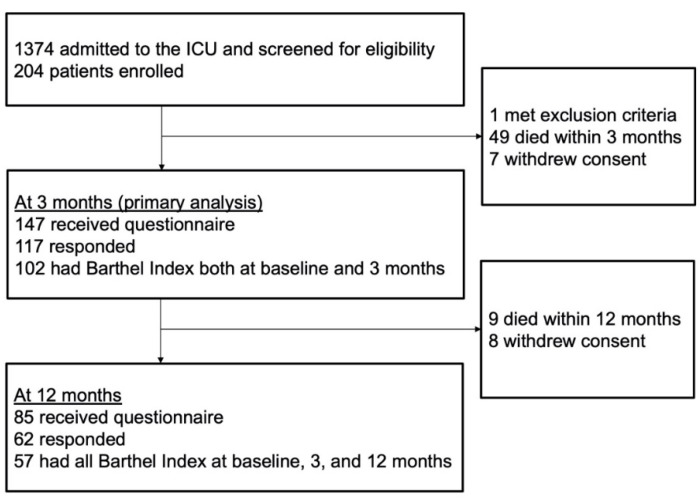
Flowchart of the patients. ICU: intensive care unit.

**Figure 2 jcm-11-01990-f002:**
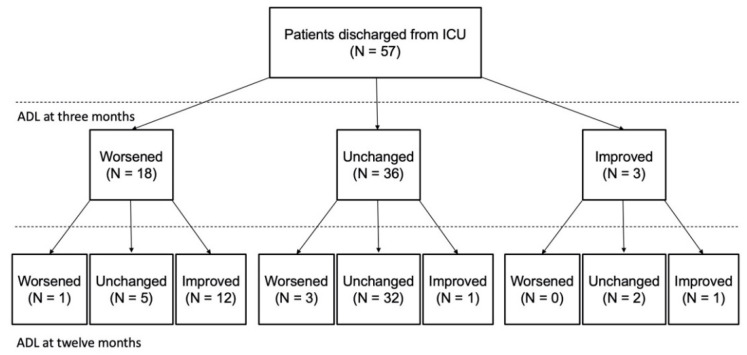
Trajectories of basic activities of daily living status within one year after intensive care unit discharge. We showed the trajectories of basic activities of daily living status (ADL) at three and twelve months after intensive care unit discharge. Worsened ADL was defined as a ≥10 decrease in Barthel index, and improved ADL was defined as a ≥10 increase in Barthel index from baseline to three months, or from three months to twelve months. The remainder of the patients were classified as unchanged. ICU: intensive care units; ADL: activities of daily living.

**Table 1 jcm-11-01990-t001:** Patient characteristics.

Characteristics	All Analyzed Patients (*n* = 102)	Patients Had Worsened ADL (*n* = 42)	Patients Did Not Have Worsened ADL (*n* = 60)	*p* Value
Characteristics at baseline				
Age, y, median (IQR)	72 (60–79)	77 (67–84)	68 (57–75)	0.0016
Male, *n* (%)	56 (55)	23 (55)	33 (55)	1.00
APACHE II score at ICU admission, mean ± SD	20.0 ± 5.9	20.8 ± 5.1	19.4 ± 6.3	0.25
Charlson comorbidity index, median (IQR)	1 (0–2)	2 (0–3)	1 (0–1)	0.012
Barthel index (BI) prior to ICU admission, median (IQR)	100 (79–100)	100 (59–100)	100 (93–100)	0.022
ADL disability (BI ≤ 60) prior to ICU admission, *n* (%)	19 (19)	12 (29)	7 (12)	0.040
Mental illness prior to ICU admission, *n* (%) ^1^	12 (12)	5 (12)	7 (12)	1.00
Postoperative admission, *n* (%)	29 (28)	13 (31)	16 (27)	0.66
Reason for ICU admission				0.22
Sepsis, *n* (%)	42 (41)	18 (43)	24 (40)	
Trauma, *n* (%)	19 (19)	11 (26)	8 (13)	
Others, *n* (%)	41 (40)	13 (31)	28 (47)	
Admission route				0.50
Emergency department, *n* (%)	70 (69)	27 (64)	43 (72)	
Operating room, *n* (%)	31 (30)	15 (36)	16 (27)	
General wards, *n* (%)	1 (1)	0 (0)	1 (2)	
Patients with disease that could directly influence brain function, *n* (%) ^2^	14 (14)	6 (14)	8 (13)	1.00
Characteristics during ICU				
Patients received mechanical ventilation during ICU stay, *n* (%)	89 (87)	37 (88)	52 (87)	1.00
Ventilator days in ICU, median (IQR) ^3^	4 (2–7)	5 (3–9)	3 (2–7)	0.024
Vasopressor therapy during ICU, *n* (%)	62 (61)	22 (52)	40 (67)	0.16
Renal replacement therapy during ICU, *n* (%)	12 (12)	4 (10)	8 (13)	0.76
Early rehabilitation during ICU, *n* (%) ^4^	70 (69)	30 (71)	40 (67)	0.67
Early enteral nutrition during ICU, *n* (%) ^4^	46 (45)	19 (45)	27 (45)	1.00
Length of sedatives/analgesics exposure				
Propofol, d, median (IQR)	2 (0–4)	2 (1–4)	2 (0–3)	0.17
Dexmedetomidine, d, median (IQR)	2 (0–4)	2 (0–4)	2 (0–5)	0.44
Midazolam, d, median (IQR)	0 (0–0)	0 (0–0)	0 (0–0)	0.43
Fentanyl, d, median (IQR)	3 (2–6)	3 (2–6)	3 (2–6)	0.69
Delirium in ICU, *n* (%) ^5^	29 (28)	16 (38)	13 (22)	0.079
Length of ICU stay, median (IQR)	5 (3–8)	6 (3–9)	5 (3–8)	0.16
ADL status at three months				
BI at three months, median (IQR)	90 (40–100)	35 (5–76)	100 (96–100)	<0.0001
Change in BI between baseline and three months, median (IQR)	0 (−20–0)	−25 (−65–−15)	0 (0–0)	<0.0001

^1^ Mental illness was defined if the medical institutions had diagnosed it or had prescribed any drugs for it. ^2^ Diseases that directly influence brain function include cardiopulmonary arrest, traumatic brain injury, stroke, and acute poisoning. ^3^ Ventilator days were calculated after excluding patients who did not receive mechanical ventilation during ICU stay. ^4^ Early rehabilitation and early enteral nutrition were defined as patients that received these therapies within 48 h from ICU admission. ^5^ Delirium was defined as at least one positive confusion assessment method for the intensive care unit during their ICU stay. IQR: interquartile range; APACHE II: Acute Physiology and Chronic Health Evaluation II; ICU: intensive care unit; SD: standard deviation; BI: Barthel index; ADL: activity of daily living.

**Table 2 jcm-11-01990-t002:** Predictors for worsened ADL status at three months after intensive care unit discharge.

	Univariable or (95% CI)	*p* Value	Multivariable or (95% CI)	*p* Value
Characteristics at baseline				
Age > 70 years old	3.35 (1.45–7.70)	0.0045	3.68 (1.33–10.19)	0.012
Male	1.01 (0.46–2.23)	0.98		
APACHE II score at ICU admission	1.04 (0.97–1.11)	0.25		
High burden of chronic illness (Charlson comorbidity index >1)	3.62 (1.53–8.56)	0.0035	4.11 (1.43–11.81)	0.0087
ADL disability (Barthel index ≤60) prior to ICU admission	3.03 (1.08–8.52)	0.036	1.31 (0.37–4.57)	0.67
Mental illness prior to ICU admission ^1^	1.02 (0.30–3.47)	0.97		
Post-operative admission category (non-post-operative admission as reference)	1.23 (0.52–2.94)	0.64		
Sepsis as reason for ICU admission (others as reference)	1.13 (0.51–2.50)	0.77		
Disease that could directly influence brain function ^2^	1.08 (0.35–3.39)	0.89		
Characteristics during ICU				
Four or more days of mechanical ventilation received	2.14 (0.96–4.79)	0.062	2.83 (1.04–7.69)	0.041
Vasopressor therapy received	0.55 (0.24–1.24)	0.15	0.39 (0.15–1.06)	0.065
Renal replacement therapy received	0.68 (0.19–2.44)	0.56		
Early rehabilitation ^3^	1.25 (0.53–2.95)	0.61		
Early enteral nutrition ^3^	1.01 (0.46–2.23)	0.98		
Delirium ^4^	2.22 (0.93–5.34)	0.073	1.95 (0.71–5.43)	0.20
Five or more days of ICU length of stay	1.68 (0.75–3.78)	0.21		

^1^ Mental illness was defined if the medical institutions had diagnosed it or had prescribed any drugs for it. ^2^ Diseases that directly influence brain function include cardiopulmonary arrest, traumatic brain injury, stroke, and acute poisoning. ^3^ Early rehabilitation and early enteral nutrition were defined as patients that received these therapies within 48 h from ICU admission. ^4^ Delirium was defined as at least one positive Confusion Assessment Method for the Intensive Care Unit during their ICU stay. OR: odds ratio; APACHE II: Acute Physiology and Chronic Health Evaluation II; ICU: intensive care unit; ADL: activity of daily living.

## Data Availability

The data that support the findings of this study are available from the corresponding author upon reasonable request.

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
