# Peer review of "Incidence and Risk Factors of Worsened Activities of Daily Living Status Three Months after Intensive Care Unit Discharge among Critically Ill Patients: A Prospective Cohort Study"

_jcm, 2022, doi:10.3390/jcm11071990_

Round 1

Reviewer 1 Report

In this article the authors study a very important point of the post intensive care syndrome: the recovery and how to evaluate it. The functional evaluation is performed using the Barthel index to assess the lack of activity of daily living 3 months after discharge from the intensive care unit. To carry out this analysis, the authors selected among the patients from the Wakayama Post-Intensive Care Syndrome study those for whom they had the Barthel index measurement before admission to the ICU (baseline) and at 3 months (n=102). The existence of an analysis of the functional status of patients before their stay in the ICU is an important strength of the study, but I have several comments:

Major comments:

1/ The primary endpoint of the study is the comparison of the Barthel index before admission to the ICU with the same measurement 3 months after discharge from the ICU. It may seem surprising to think that an ICU stay could improve the patient's functional status? Isn't it the improvement of the Barthel index between discharge from the ICU and 3 months later?

2/ Still concerning the evaluation of the primary endpoint, the improvement of the Barthel index is considered significant if the index increases by more than 10. This value seems to come from an article by Hsieh YW et al. published in 2007 (reference 11). However, this reference mentions a clinically significant value of 1.85 to judge the evolution of the Barthel index. How did the authors choose the value of 10?

Minor comments:

1/ The presentation of table 1 on 2 different pages does not promote readability. Moreover, the layout of the table header is not aligned with the columns, which may confuse the interpretation.

2/ In order to situate the interest for the intensive care patient population, it could be interesting to have the initial number of patients admitted over the period. What percentage do these 204 initial patients represent?

3/ The sedations used during the ICU stay are likely to influence the occurrence of delirium and possibly the PICS. We have no information on the nature and duration of the sedative drugs used.

Author Response

Thank you very much for your review and constructive comments. According to your comments, we modified the manuscript. We have highlighted the changes made to the manuscript in yellow.

You can find the detailed reply for your comment in the attached file.

Reviewer 2 Report

Greetings

I applaud the work. The topic is relevant. Data collection, analysis, and manuscript writing are good and limitations are well noted. I do not have any major comments to make. However, in my opinion, if you include the following points in the results and discussion, it will be better (more informative)

  1. What about the cases who might have developed some infections or other morbidities unrelated to the disease which led to ICU admission? Were they excluded or considered?
  2. It will be interesting to know the diagnosis of the patients and the nature of the disease (chronic or acute) in the patients who had decreased ADL. 
  3.  Was there a difference between the decreased ADL cohort and ADL static or increased group in terms of chronicity (acute or chronic illness) of the disease which led to emergent ICU admission?

Best of luck

Author Response

(The authors gave the same response as above.)

Round 2

Reviewer 1 Report

For information, in my pdf version of the manuscript, the column headings of table 1 still do not appear very clearly above the columns of numbers.